# Co-Occurrence of *L. monocytogenes* with Other Bacterial Genera and Bacterial Diversity on Cleaned Conveyor Surfaces in a Swine Slaughterhouse

**DOI:** 10.3390/microorganisms10030613

**Published:** 2022-03-14

**Authors:** Tamazight Cherifi, Julie Arsenault, Sylvain Quessy, Philippe Fravalo

**Affiliations:** 1Chaire de Recherche en Salubrité des Viandes, Faculté de Médecine Vétérinaire, Université de Montréal, Saint-Hyacinthe, QC J2S 2M2, Canada; ctamazight@gmail.com (T.C.); sylvain.quessy@umontreal.ca (S.Q.); 2Centre de Recherche en Infectiologie Porcine et Avicole (CRIPA-FQRNT), Faculté de Médecine Vétérinaire, Université de Montréal, Saint-Hyacinthe, QC J2S 2M2, Canada; julie.arsenault@umontreal.ca; 3Groupe de Recherche et d’Enseignement en Salubrité des Aliments (GRESA), Faculté de Médecine Vétérinaire, Université de Montréal, Saint-Hyacinthe, QC J2S 2M2, Canada; 4Epidemiology of Zoonoses and Public Health Research Unit (GREZOSP), Faculté de Médecine Vétérinaire, Université de Montréal, Saint-Hyacinthe, QC J2S 2M2, Canada; 5Pôle Agroalimentaire du Cnam, 22440 Ploufragan, France

**Keywords:** *Listeria monocytogenes*, microbiota, swine slaughterhouse

## Abstract

Bacterial pathogens, such as *Listeria monocytogenes*, can show resistance to disinfection and persistence on working surfaces, permitting them to survive and contaminate food products. Persistence—a complex phenomenon involving interactions between many bacteria within a biofilm—is modulated by in situ characteristics. This study aimed to describe, in silico, the microbiota identified in a swine slaughterhouse after sanitation procedures to better understand the presence of *L. monocytogenes* on these surfaces. Molecular tools for characterization of microbial communities were used to assess the relative contribution of different bacteria resulting from this phenomenon, and the 16S rRNA sequencing method was used on samples from meat conveyor belt surfaces collected on four sampling visits to study the co-occurrence between *L. monocytogenes* and other bacteria. From the background microbiota, a total of six genera were found to be negatively correlated with *Listeria* spp., suggesting *Listeria* growth inhibition, competition, or at least an absence of shared habitats. Based on these results, a complete scenario of interactions of *Listeria* with components of background microbiota was established. This work contributes to identifying avenues that could prevent the growth and persistence of *L. monocytogenes* on food-processing surfaces.

## 1. Introduction

The presence of *Listeria monocytogenes* in food processing plants is a major concern for industries and food control authorities [1,2,3]. The ubiquitous nature of this bacterium combined with its ability to grow in harsh conditions, including high salt concentrations, high acidity levels, and large temperature ranges, makes its control within food production environments particularly challenging [4]. Indeed, the ability of *L. monocytogenes* to persist within food-processing environments was reported in many studies [4,5,6,7], and the circumstances associated with this persistence phenomenon appear to be complex [8]. *L. monocytogenes*’ biofilm formation and resistance to commonly used disinfectants in food industries may partly explain this persistence [5,6,7,9]. Some authors found an increased ability for biofilm formation in *L. monocytogenes*’ persistent strains. Moreover, the detection of genes conferring resistance to disinfectants, such as quaternary ammonium compounds (QACs), was correlated with persistence phenotypes in strains isolated from food processing plants [2,10,11]. Another factor that could influence the persistence of *L. monocytogenes* in the food industry is the structure and composition of bacterial communities present in biofilms, which are expected to include species other than *L. monocytogenes* [12,13]. As an example, it was reported that microbial communities dominated by Gram-negative bacterial species identified on a salmon slaughterhouse conveyor belt seemed to have an inhibitory effect on *L. monocytogenes* in a mixed biofilm [13]. This previous study was based on a culture-based method to identify communities present on conveyor surfaces. Consequently, only the most abundant and cultivable bacteria were detected, which limited the ability to explore links and/or interactions between *L. monocytogenes* and other bacteria present on biofilms from meat conveyor surfaces. While bacteria from specific species could influence the occurrence of other species for many reasons—such as competing for the same resources [14,15]—the identification of co-occurrences between *L. monocytogenes* and other species might help us find innovative solutions for controlling the proliferation and/or persistence of this pathogen.

So far, there is scant information about the co-occurrence of *L. monocytogenes* and other bacteria species over time in the food production environment. Studies that were conducted include one that showed *L. monocytogenes* was outcompeted by a mixture of bacterial species present on wooden shelves mainly due to competition for nutrients [16,17]. Inhibition of *L. monocytogenes* growth could also be the result of bacteriocin secretion by other species, such as *Enterococcus faecium* [18]. Other studies, in contrast, demonstrated positive interactions between *L. monocytogenes* and other bacterial genera [19,20]. Indeed, for example, *Flavobacterium* spp., which was part of the background microbiota of a seafood processing plant, was reported to enhance the colonization of *L. monocytogenes* on stainless steel surfaces [19]. Finally, others reported no interaction between *L. monocytogenes* and other bacteria such as *Pseudomonas fluorescens* [12,21].

The aim of this study was to describe, independent of the cultivability of the bacteria, (1) the microbiota diversity on meat conveyor surfaces from a swine slaughterhouse after cleaning and disinfection procedures and (2) the co-occurrence of *Listeria* spp. with other bacterial communities present on meat conveyor surfaces.

## 2. Materials and Methods

### 2.1. Sample Collection, Processing, and Culture of Listeria monocytogenes

Samples were collected in the cutting facility of one swine slaughterhouse during four visits and after cleaning and disinfection operations. The disinfection procedures were applied by industry employees as part of their routine activities and consisted of an application of QACs-based disinfectants at a concentration between 150 and 200 ppm, on a daily basis, after each meat-cutting process.

On each visit, the belt of three meat conveyors was sampled at the beginning, middle, and end of the conveyor. Each sampling consisted of a swabbing of 1 m^2^ of the belt surface after mechanical mobilization (brushing) using prewarmed wet swabs with a D/E (Dey-Engley) neutralizing broth (Innovation Diagnostic, Saint-Eustache, QC, Canada) to neutralize a broad spectrum of disinfectants and antiseptics. A total of 48 swab samples were collected. Each swab was shaken vigorously in 100 mL of sodium chloride solution (0.9% of NaCl in nuclease-free water). A volume of 10 mL was transferred into a separate tube and centrifuged, and the supernatants were discarded. The tubes containing pellets were immediately stored at −80 °C for DNA extraction. The detection of *L. monocytogenes* was performed on the 90 mL of remaining suspension according to the Compendium of Analytical Methods, MFHPB-30 [22] with few modifications. Briefly, all samples were enriched in 90 mL of 2X concentrated University of Vermont media 1 (UVM-1; Innovation Diagnostics, Saint-Eustache, QC, Canada) and incubated at 30 °C for 48 h. A second enrichment with Fraser broth media (Innovation Diagnostics, Saint-Eustache, QC, Canada) was used at 37 °C for 24 h. The cultures were plated on a selective COMPASS *Listeria* agar (Innovation Diagnostics, Saint-Eustache, QC, Canada) media, and typical colonies were streaked on sheep blood agar (Oxoid, Nepean, ON, Canada). The confirmation of *Listeria monocytogenes* was performed by a polymerase chain reaction (PCR) used for serogrouping [23]. In addition to these swab samples, one control, consisting of a clean swab and brush transported during the sampling visit, was collected on two sampling visits (1 and 4) and submitted to the same bacteriological and molecular analyses as the swab samples.

### 2.2. DNA Extraction, 16S rDNA Construction Library, and Bioinformatics Analysis

The conserved pellets from the step above were used for phenol chloroform DNA extraction protocol. Briefly, bacteria were lysed with a lysis buffer (Tris-Hcl, EDTA, NaCl, and SDS) and content was extracted using glass beads by vortexing using the FastPrep procedure (MP Biomedical, Solon, OH, USA). The phenol chloroform isoamylic (25:24:1 *v*:*v*) solution was added to the supernatant. After mixing by inversion for 2 min, the solution was centrifuged, and the pellets were discarded. An additional 2 min of mixing was conducted, and, finally, the solution was centrifuged to collect the supernatant. Ammonium acetate (10 mM) and cold ethanol 100% were added to the supernatant, and the suspension was kept at −80 °C for 24 h for DNA precipitation. After precipitation, the tubes were centrifuged at 21,004× *g* for 15 min, and the pellets were washed with ethanol before drying. The DNA was solubilized in 50 µL of nuclease-free water and stored at −20 °C before PCR amplification of the 16S rDNA V4 region. The primers 515F/806R pair [24] was used to amplify the V4 region, and the libraries were prepared using the NEXTERA kit (Illumina, Inc., San Diego, CA, USA) following recommendations from the manufacturer. The Illumina Miseq technology was used to perform the paired-end sequencing at Genome Québec Innovation Center (https://www.genomequebec.com/en/home/, accessed on 12 April 2016).

Data were analyzed using Quantitative Insights Into Microbial Ecology (QIIME) 2 v 2017.12 pipeline [25]. The demultiplexed reads were filtered and denoised using Dada2 v 2017.12.1 software [26]. The first 13 low-quality bases and the last 10 ones (at position 240) from both of the reverse (R1) and forward (F1) reads were trimmed during this step based on the quality plots that QIIME 2 generated. The resulting two separated tables containing amplicon single variants (ASVs) and sequences of each ASV were used for the downstream analyses.

ASVs table and their corresponding sequences were clustered into operational taxonomic units (OTU) using VSEARCH [27] v2018.8.0 and a closed-reference clustering with identity cutoff set at 99% against the SILVA database [28]. To minimize the impact of potential false positives due to cross-contamination, we used the decontam R package [29] to detect and remove potential OTUs detected as contaminants, using default settings.

Taxonomic composition of samples was determined by performing a pre-trained, naïve Bayes classifier on the reference taxonomy and sequences from SILVA database with 99% identity [28]. The feature classifier sklearn-classify [28] was used for taxonomic classification.

### 2.3. Diversity of OTUs by Sampling Visit and Listeria monocytogenes Culture Status

The alpha and beta diversities of OTUs were estimated for each sample rarefied at 20,000 sequences per sample. This threshold was defined based on the sample having the lowest number of sequences (*n* = 20,287) and was supported by the exploration of the rarefaction curves performed using phyloseq R package [30] v1.26.1, considering that this sample size was sufficient to reach the curve plateau for the majority of samples (Appendix A).

The alpha diversity, Shannon diversity index [31], and Shannon evenness index [32] were estimated in each rarefied sample. For the two indices, the median values of samples were compared between visits using a Kruskal–Wallis test followed by pairwise comparisons with Bonferroni adjustment for multiple testing. The median values were also compared according to the *L. monocytogenes* culture status of samples using a Wilcoxon test.

For the beta diversity, Bray–Curtis and Jaccard dissimilarity indices were estimated to explore the quantitative (community abundance) and qualitative (presence/absence) dissimilarities of OTUs in each sample, respectively, using phyloseq package in R. The Bray–Curtis and Jaccard indices were compared between visits and according to *L. monocytogenes* culture status using pairwise PERMANOVA analyses with 999 permutations.

### 2.4. Description of the Taxonomy Composition

To explore the taxonomy composition of the bacterial communities in OTUs from sampled conveyor surfaces, the relative abundance of their genera was described using the phyloseq package. All OTUs with missing values for the genus rank were excluded. The relative abundance of the most frequent genus in OTUs according to the sampling visits and to the *L. monocytogenes* culture status was also evaluated.

### 2.5. Description of Bacterial Communities

To explore the relationships between microbial communities, particularly between *Listeria* and other genera, a network was created from positive and/or negative correlations between bacterial genera. Rare genera were first filtered to remove all OTUs corresponding to the genus with fewer than 20 occurrences across all samples. The filtered table was then used to construct the network based on three indicators: Spearman correlation, Bray–Curtis, and Kullback–Leibler dissimilarities. This network was built using the CoNet program [33] and implemented with Cytoscape software v3.3.0 [34]. A threshold of 1500 top- and bottom-scoring (for anti-correlations) links was included for each of the three indicators. For each link between two genera, 100 renormalized permutations and bootstrap score distributions were computed. Brown’s method was used to merge all measure-specific *p*-values of the indicators [35], with false discovery rate controlled at 5% using the Benjamini–Hochberg correction. Only links with confirmed associations for all 3 indicators were kept in the final network. The resulting network was visualized in Gephi v 0.9.2 [36] using the Yifan Hu algorithm [37]. Nodes (i.e., genera with significant interactions) were clustered using the constant Potts model with the resolution set at 0.1 [38] to identify the community structure of the network, with the size of each node representing the relative abundance of the genus. A second network was created from the previous one limited to the nodes directly or indirectly connected to *Listeria* spp. For better clarity of the visualization, the network was filtered based on the Spearman correlation coefficient used in the analysis as the weight of the link. Thus, only nodes with a correlation coefficient higher than 0.45 were kept.

## 3. Results

*L. monocytogenes* was detected by a culture-based method in 13 (27.1%) out of the 48 samples. The first and second visits showed the highest proportion of positive samples (Table 1).

From the 48 samples, three samples were not sequenced due to the small amount of extracted DNA. The number of sequences in the remaining 45 samples after quality control procedures using Dada2 ranged from 20,287 to 229,410 sequences per sample (Appendix A). All downstream analyses were performed based on the metadata table shown in Appendix A.

### 3.1. Diversity of the Microbiota on Conveyor Surfaces Was Different between Visits

Shannon diversity and evenness indices showed a significant difference of both richness and evenness in OTUs (Kruskal–Wallis, *p* = 0.00005 and *p* = 0.00009, respectively) between sampling visits. Samples from visit S1 were less diverse compared to all other sampling visits (Kruskal–Wallis, *p* < 0.05, Figure 1a). Moreover, a significantly higher diversity and higher evenness were observed in S2 compared to S4 (Figure 1a). Alpha diversity comparisons between culture-positive and culture-negative samples for *L. monocytogenes* showed no significant differences (Figure 1b).

When richness estimations were compared between visits by pairwise PERMANOVA analysis based on Bray–Curtis and Jaccard indices, the diversities were significantly different between all pairs of visits (pseudo-F test, *p* = 0.001 for both indices), except between S2 and S3 (Table 2; Figure 2a,b). Diversity comparisons between the group of samples positive and negative for *L. monocytogenes* showed no difference (Figure 2c,d).

### 3.2. Relative Abundance and Microbial Composition Changed between Visits

A total of 114 different genera were obtained from OTUs. Overall, 98.8% of the OTUs were from the five most abundant phyla, i.e., *Proteobacteria*, *Actinobacteria*, *Bacteroidetes*, *Firmicutes*, and *Acidobacteria*. The most frequent phylum was *Proteobacteria* with a relative abundance of 73.2%, followed by *Bacteroidetes* (12.4%), *Actinobacteria* (9.9%), and *Firmicutes* (2.7%). *Pseudomonas*, *Sphingomonas*, *Acinetobacter*, and *Chryseobacterium* were the most abundant genera identified. Proportions of OTUs belonging to these genera changed between sampling visits. *Pseudomonas* was highly dominant in S1, whereas the dominant genus was *Acinetobacter* in S2 and S3 and *Sphingomonas* in S4 (Figure 3a). *Listeria* appeared among the least abundant genera; thus, it was identified almost exclusively in S1 and S2, which were also the visits with the highest proportion of culture-positive *L. monocytogenes* samples. Results showed that *Pseudomonas* was the most abundant genus in OTUs from *L. monocytogenes* culture-positive samples, whereas *Sphingomonas* was the most abundant genus in the culture-negative samples (Figure 3b).

### 3.3. Differential Community Interaction

Our results showed the presence of correlations forming 1027 links and involving 90 nodes representing the genera before applying the threshold of 0.45 for the Spearman correlation coefficient (Figure 4a and Appendix A). A total of 46 genera interacted with *Listeria* negatively and interestingly; *Pseudomonas* was observed to interact negatively with the genus *Williamsia*, and this latter had a direct link with *Listeria* (Figure 4a).

When a threshold of ≥0.45 for the Spearman correlation was applied from the previous step and the network was filtered to keep only genera interacting directly or indirectly with *Listeria* genus, one community was found to interact with *Listeria* community, which was composed of 13 other genera, covering 21.1% of the nodes (Figure 4b). According to our results, the *Listeria* genus had no direct positive correlation with any other bacterial genera (Figure 4b), but direct negative correlations were present between *Listeria* and the six genera, *Herminiimonas*, *Bryobacter*, *Caulobacter*, *None* (genus not identified), *Sphingomonas*, and *Mycobacterium*, belonging to the six phyla, *Proteobacteria*, *Acidobacteria*, *Proteobacteria*, *Elusimicrobia*, *Proteobacteria*, and *Actinobacteria*, respectively (Figure 4b). Interestingly, *Proteobacteria* and *Actinobacteria* were the phyla which had the highest numbers of direct associations with *Listeria*, while *Acidobacteria*, *Bacteroidetes*, and *Elucimicrobia* had only two, one, and one genus, respectively, interacting with *Listeria*. From the *Proteobacteria* phylum, the *Sphyngomonas* genus accounted for the most abundant genera interacting negatively with *Listeria* (Figure 4b).

## 4. Discussion

In this study, we described the diversity of the microbiota present on swine meat conveyor surfaces from four sampling visits. We also identified, in silico, the co-occurrence of this microbiota with the *Listeria* genus, including *Listeria monocytogenes*, as confirmed by a culture-based method.

The proportion of culture-positive samples for *L. monocytogenes* observed in this study was in the same range as reported in other studies [4]. The relative abundance of *Listeria* spp. revealed by 16S metagenomic in culture-positive samples for *L. monocytogenes* was higher than in culture-negative samples. This concordance between the two methods may suggest that *L. monocytogenes* was the most prevalent species among *Listeria* spp. present on conveyor surfaces. Moreover, classification at the species level only identified *L. monocytogenes* among the *Listeria* genus (Appendix A).

Diversity analyses showed significant differences in both composition and organization of microbial communities depending on the sampling visits. Continuous new sources of bacteria from incoming materials or persons entering the processing environment should be considered in interpreting such diversity. It was reported, for example, that carcasses contaminated by *L. monocytogenes* from internal organs, such as tonsils and tongue, during the slaughtering process may contaminate plant environments [39,40]. Additionally, the plant workers can be an important source of contamination of food processing plant environments and products since humans are known to be an important reservoir, and the risk of contamination is associated with the level of sanitary practices [41].

From one batch to another, the microbiota composition changed, and these modifications were particularly noticeable due to the long intervals of time in our study (i.e., from three weeks to eight months between two consecutive visits). In addition, fluctuations in humidity, temperature, nutrient access, shear forces, and other physicochemical stresses could affect microbiota composition on conveyor surfaces over time, and one may think that only bacteria that can withstand these fluctuations could establish a resident niche, as supported by Fagerlund, Møretrø, Heir, Briandet, and Langsrud [12].

Overall, there was no significant difference in diversity indices of background microbiota between the culture-positive and culture-negative samples and *L. monocytogenes*. In our conditions, the presence of *L. monocytogenes* was not affected by the variability (relative abundance of species) and variety (number of different species) of species composing the microbiota of the meat conveyor surfaces.

According to taxonomic classification, *Proteobacteria* was the most abundant phylum, followed by *Bacteroidetes*, *Actinobacteria*, *Firmicutes*, and *Acidobacteria*. Few data are available to confirm these distributions. Despite most phyla being the same as observed in another study, *Actinobacteria* was reported as one of the most abundant phyla detected in meat samples [42]. This discrepancy could be due to their sampling period, which was before cleaning and sanitation on meat conveyor surfaces, whereas our sampling was done after cleaning and sanitation. It was previously reported that such factors may affect the composition of the bacterial community [12].

Among the 114 identified genera, the Gram-negative *Sphingomonas*, *Pseudomonas*, *Acinetobacter, Chryseobacterium*, and *Caulobacter* were the most abundant ones across all sampling visits. These genera have already been pointed out as the most abundant background microbiota of meat conveyor surfaces [12], as well as of processing plant equipment in fish production [2,12,13]. Furthermore, *Pseudomonas* and *Acinetobacter* were reported previously for their ability to survive cleaning and their high biofilm-forming ability [12], which may help to explain their high abundance. The *Sphingomonas* genus was not detected when culture-based approaches were used, while sequence-based, cultivation-independent approaches found this genus as the most abundant, with *Pseudomonas* spp. [43]. The failure to detect *Sphingomonas* with a culture-based method may be the result of higher requirements in growth factors, such as Mg^2+^ for some *Sphingomonas*, as suggested by some researchers [44], and/or to interspecies competition [43]. Our findings confirmed this genus forms a significant part of the microbial communities and underlined the complementarity of both methods for description of bacterial composition in a given environment.

*Listeria* and *Staphylococcus* were among the most abundant genera in the *Firmicutes* phylum detected on these cleaned meat conveyor surfaces. Considering previous studies describing *Listeria* (and particularly *L. innocua* and *L. monocytogenes*) as a weak competitor in presence of *Pseudomonas* and *Acinetobacter* [12,13], it was expected that this genus would be less abundant in our communities, particularly when both *Pseudomonas* and *Acinetobacter* were well represented. The co-occurrence of *Listeria* and *Pseudomonas* genera reported in this study has to then be considered in accordance with other studies where an increase in the number of cells of *L. monocytogenes* was reported with the growth of *Pseudomonas*
*putida* or *P. fragi* biofilms [12,45]. Moreover, when *Pseudomonas* and *Listeria* were grown in a mixed biofilm with other genera, it was shown that *Pseudomonas* dominated but did not eliminate *Listeria* from the biofilm [13]. Our results suggest that *Listeria* formed niches within *Pseudomonas* biofilm and could withstand cleaning and disinfection procedures and other harsh conditions. All the aforementioned statements were, however, based on culture-based methods, which are limited for identifying several bacteria from a given sample. Indeed, the strong competition that occurs during enrichment steps in cultures, as well as the viable but non-cultivable properties of some bacterial species on surfaces, underlines the need for non-culture-based analysis. For this reason, we aimed to study these interactions considering all bacterial populations using the 16S rDNA sequence-based methods. It is worth noting that these methods cannot distinguish between dead and viable bacteria. Thus, once possible associations are identified, the interactions might need to be confirmed, e.g., via cultivation-based methods.

At a community network level, considering all bacterial populations and based on correlation and dissimilar methods, only negative correlations between *Listeria* and other genera were detected. Such results were previously reported where, among twenty-nine bacterial species recovered from a dairy environment, sixteen induced reduction of biofilm formation by *L. monocytogenes* [46]. Such findings suggest that these populations are not mutualistic; either they do not share similar niches or a competition exists, as reported previously [47]. The competition between species could be related to nutrient limitation and augmented by antimicrobial compound production, as reported by Giaouris et al. [9]. Negative associations were often reported with *Listeria* spp., particularly for *L. monocytogenes*, since its presence, particularly in biofilm co-culture, was repeatedly hampered [13,48,49]. *Listeria monocytogenes* is recognized as a weak competitor when placed in a co-culture [2,12,13], which is consistent with our results. It is noteworthy that, except for *Sphingomonas*, these bacteria have never been cultivated along with *Listeria* spp. The low abundance of these genera compared to *Pseudomonas* and *Sphingomonas* and the difficulty of cultivation or non-cultivable status of some of these genera, such as certain species of *Herminiimonas* [50] and *Pseudoclavibacter* [51], could be the reasons why they were not detected in classical bacteriology studies. The negative association between *Listeria* and *Sphingomonas* is supported by a previous study where *Sphingomonas* inhibited the growth of *L. monocytogenes* by production of astaxanthin [52]. Interestingly, species from the *Paracoccus* genus, such as *P. marcusii*, which have a negative interaction with *Listeria* genus, are reported to produce astaxanthin as well [53,54]. Similar, negative associations with the *Listeria* genus or *L. monocytogenes* were observed with other species such as *Lactococcus lactis* [48].

## 5. Conclusions

In summary, this study showed significant differences in composition and abundances of residual microbiota on pork meat conveyor surfaces over time, which highlights the non-stable bacterial community composition in the meat industry environment. However, the core bacteria population, represented by the most abundant genera, was often the same over time in this environment. From this background microbiota, a total of six genera, such as *Sphingomonas*, were found to interact negatively with *Listeria* spp., which could be explained by various factors, such as growth inhibition, absence of shared habitats, or competition, but further studies based on cultured methods are needed to interpret these negative interactions. These results provide interesting information regarding the relationship between background microbiota and *Listeria* spp. in order to identify potential bacterial species inhibiting its persistence within the food production environment.

## Figures and Tables

**Figure 1 microorganisms-10-00613-f001:**
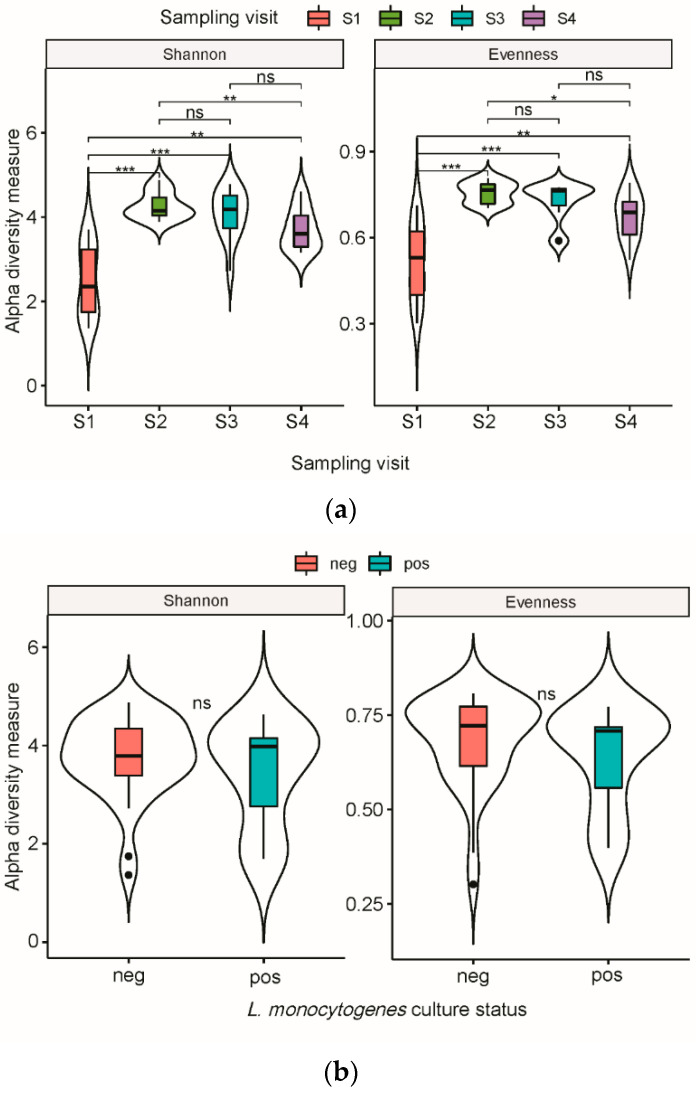
Violin box plot representation of alpha diversity of bacterial OTUs from meat conveyor surfaces in a cutting facility from a swine slaughterhouse. (**a**) Diversity was measured by Shannon diversity index and Shannon evenness index according to sampling visit, with medians compared using the Kruskal–Walis test. (**b**) Diversity was measured by Shannon diversity index and Shannon evenness index according to *Listeria monocytogenes* culture status, with medians compared using the Wilcoxon test. Statistical significance: n.s., *p* > 0.05 *, *p* < 0.05 **, *p* < 0.01 ***, *p* < 0.001.

**Figure 2 microorganisms-10-00613-f002:**
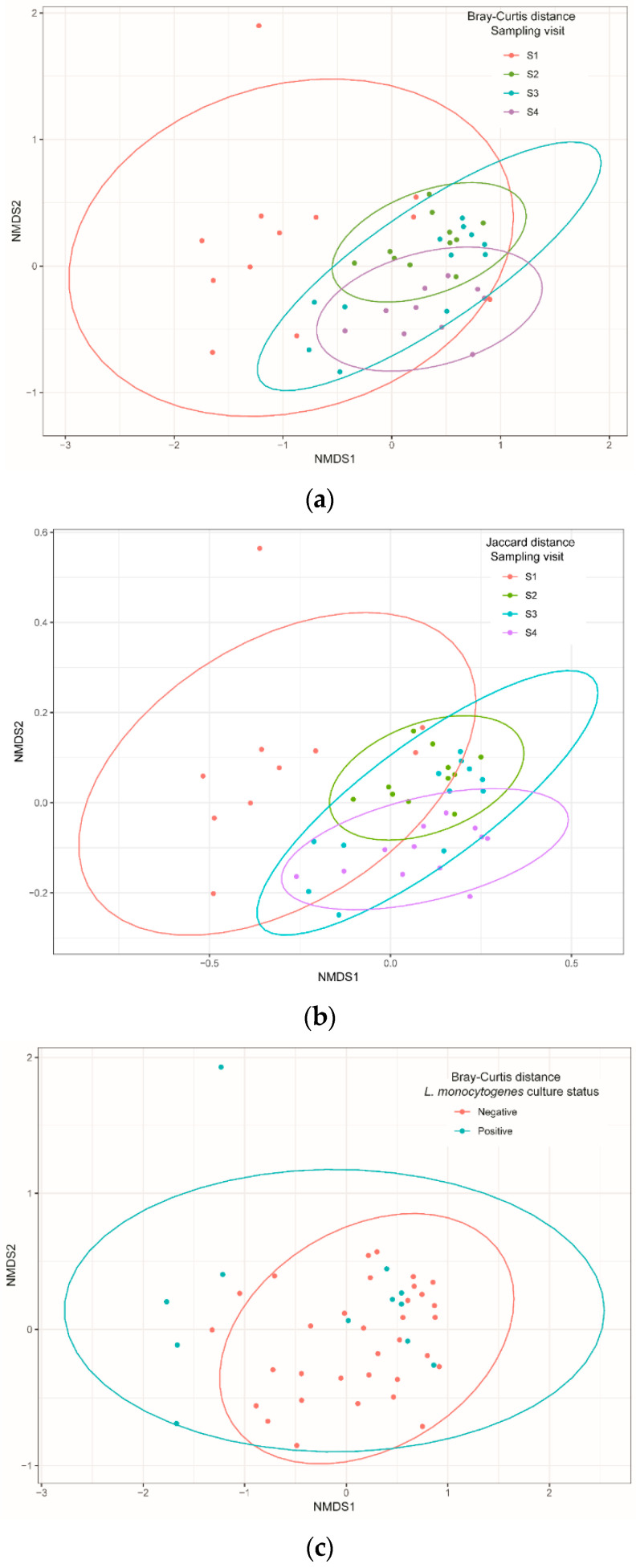
Non-metric multidimensional scaling (NMDS) of Bray–Curtis and Jaccard distances of bacteria communities identified using 16S Miseq sequencing technology on samples isolated from cutting facility conveyor surfaces according to sampling visit (**a**,**b**) and *Listeria monocytogenes* culture status (**c**,**d**). Pairwise PERMANOVA with 999 permutations (using pseudo-F ratios) was used for all comparisons.

**Figure 3 microorganisms-10-00613-f003:**
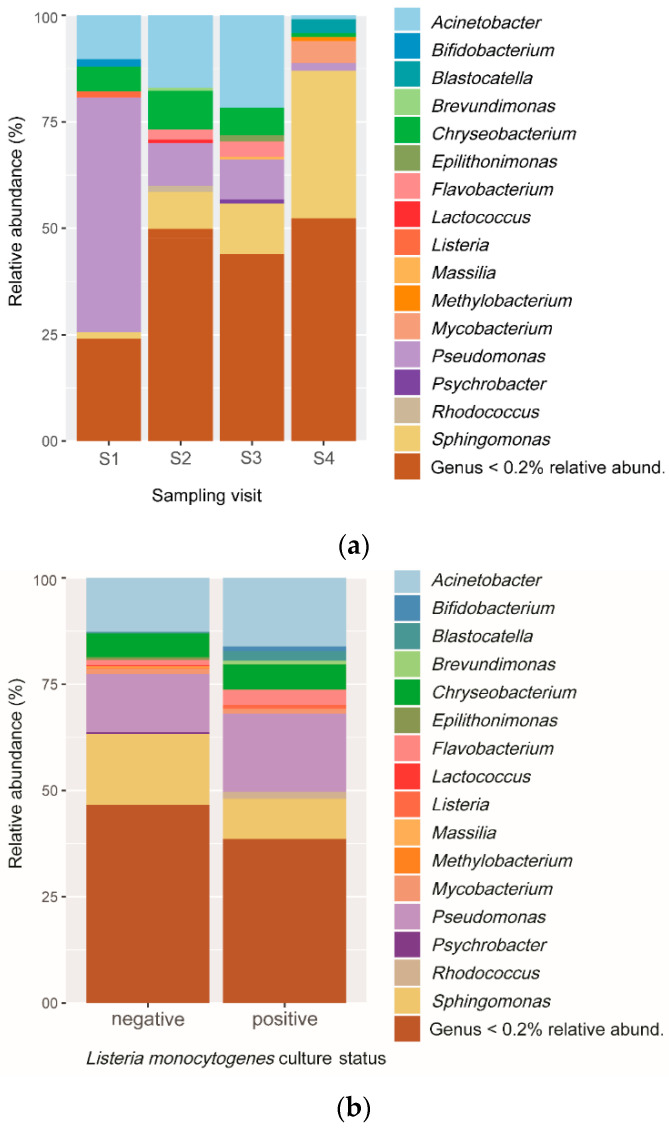
Relative frequency of different taxonomic profiles identified from OTUs in cutting facility conveyor surfaces. (**a**) Relative abundance of the top sixteen genera according to the sampling periods. (**b**) Relative abundance of the top sixteen genera according to *L. monocytogenes* culture results.

**Figure 4 microorganisms-10-00613-f004:**
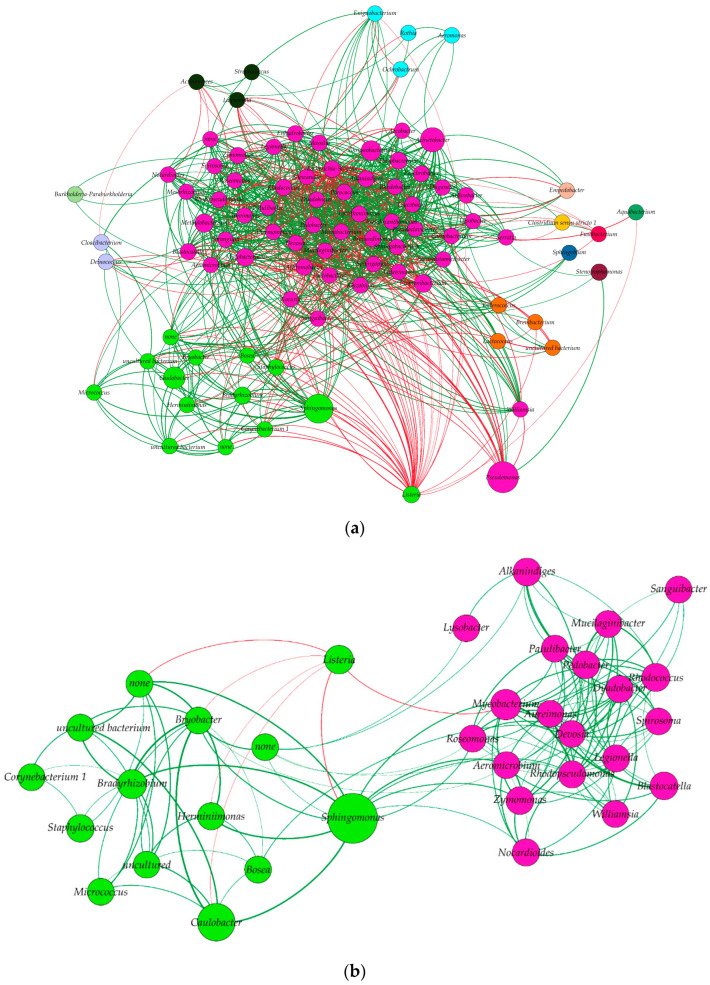
Network construction based on Spearman correlation, Bray–Curtis dissimilarities, and Kullback–Leibler dissimilarities. (**a**) General network of all associations between genera clustered according to constant Potts model (communities sharing most links between each other). The size of each node refers to the abundance of the genus, the thickness of the link refers to its weight (presented here by the Spearman correlation coefficient), and the color of each node refers to positive (green) and negative interactions (red), (**b**) Network construction based on the direct associations between *Listeri**a* genus and the other genera and indirect associations between those genera and others. Links colored in red represent negative associations between the genera while green links represent positive associations. The node labelled “none” refers to the unidentified genus from the *Elusimicrobia* phylum.

**Table 1 microorganisms-10-00613-t001:** Bacteriological detection of *L. monocytogenes* in samples from swine slaughterhouse conveyor surfaces collected after cleaning and sanitation, by sampling visit.

Sampling Visit	Date of Sampling	Number of Samples	Number (%) of Positive Samples to *L. monocytogenes* (%)
S1	23 May 2014	12 ^a^	6 (50)
S2	18 January 2015	12 ^b^	5 (41.7)
S3	7 Febrsuary 2015	12	1 (8.33)
S4	28 February 2015	12	1 (8.33)
Total		48	13 (27.1)

^a^ 2 samples were not sequenced, ^b^ one sample was not sequenced.

**Table 2 microorganisms-10-00613-t002:** Pairwise PERMANOVA analysis based on Bray–Curtis and analysis of variance using Jaccard distance matrices with 999 permutations.

Sampling Visits Compared	*p*-Value	
	Bray–Curtis	Jaccard
S1 vs. S2	0.001	0.001
S1 vs. S3	0.001	0.003
S1 vs. S4	0.001	0.001
S2 vs. S3	0.401	0.152
S2 vs. S4	0.001	0.001
S3 vs. S4	0.009	0.002

## Data Availability

Publicly available datasets were analyzed in this study. These data can be found here: PRJNA728720.

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
