# Peer review of "Co-Occurrence of L. monocytogenes with Other Bacterial Genera and Bacterial Diversity on Cleaned Conveyor Surfaces in a Swine Slaughterhouse"

_microorganisms, 2022, doi:10.3390/microorganisms10030613_

Round 1

Reviewer 1 Report

The study by Cherifi, T. et al examined the prevalence of the foodborne pathogen Listeria monocytogenes, recovered from conveyor surfaces at a swine slaughterhouse.  The present study employed high-resolution sequencing techniques (Illumina MiSeq) to conduct a metagenomic analysis and determine the positive or negative relationship between the prevalence of the detected Listeria when compared to other bacterial genera.  This study provides fundamental information that is valuable to food safety regulatory agencies as well as food processors to aid in the control of this important foodborne pathogen linked to high profile outbreaks.  The manuscript would benefit by providing more information on the sampling methods (e.g., composition of buffers used and more description on the selective/chromogenic media for isolation).

Other comments:

Page 1: The use of "culture-independent" in the title is misleading since the present study did use a selective culture-based method for defining Listeria positive samples. I suggest deleting "Culture-independent study" from the title. Start title as Co-occurence....

Lines 50-51: The sentence "However, this study..." seems out of place in the middle of this paragraph. It should be moved to the end of the intro describing what the aim of this study. Remove "however" at the beginning of the sentence.

Line 80:  Please define what is the composition of the neutralizing broth.

Line 82: Please define chemical composition of saline

Line 86:  Please define what is the MFHPB30 method. Need to describe the acronym to readers.

Line 90:  Was COMPASS Listeria agar used? Please specify.  Also COMPASS is a registered brand.

Line 91:  What kind of Oxoid blood agar was used? There are many sold by Oxoid.

Line 107:  Describe as xg not rpm so that methods can be easily transferred among different labs.

Line 111:  Suggest stating “following recommendations from the manufacturer” or “or following manufacturer's procedures.

Lines 122-123:  Please capital letters for both “vsearch” and “silva”.

Line 121-129:  These paragraphs should be combined with the previous one in lines 114-120 on bioinformatics analyses.

Line 243:  Please explain why referring to "almost exclusively in S1" when Table 1 is showing a 41.7% Listeria positive samples for S2.

Lines 279-288:  Figure needs to be higher resolution since it is difficult to read the genus. The network schematic shown in Figure 4b is more useful to readers; the one shown in Figure 4a is not possible to be interpreted.  Still the resolution and font size should be improved. Also, those genera on top of a black/dark colored circle cannot be completely read.

Lines 303-304:  Please explain how individuals can affect the microbial community in a processing plant. Provide specific evidence for this statement.

Lines 304-305:  What is known about the microbial community of swine carcasses? Please explain.

Line 308-309: Is there evidence for these fluctuations in the processing plant examined in the present study? Such information must be available. If not, then revise the statement in the manuscript.

Lines 351-352:  In this study, no evidence was provided to support the speculative argument on the role of biofilms and bacterial niches. Please revise statement in the manuscript.

Author Response

We would like to thank you for your time and constructive comments, which definitely help to improve the quality of our manuscript.

Please see the attachment for our responses.

Reviewer 2 Report

The authors used 16s RNA sequencing to identify populations of bacteria in L. monocytogenes culture-positive and culture-negative samples collected at 4 different times from conveyor surfaces in a swine slaughterhouse. Findings include that microbial diversity and evenness varied over time, but not between L. monocytogenes culture-positive and culture-negative samples. Importantly, analysis of networks linking various genera showed no positive direct interactions between Listeria and other genera, but there were negative direct interactions with 6 genera suggesting competition for niches, direct antagonism, etc. among these genera. The authors also state that these results suggest Listeria could from niches within Pseudomonas biofilms and as such withstand decontamination procedures.  If true, this would be an important finding that may indicate elimination of Listeria from food processing areas depends upon elimination of other genera that protect it.

In general the manuscript is well-written and the genomics and analyses delineated with care. The description and depiction in data in Fig 4b of interactions between Listeria and other genera, and the implications thereof need to be represented and written more clearly.

Figure 4b is a key figure in this manuscript as it shows only negative interactions between Listeria and other genera. As such the description of it in 261-270 could be improved. For example, the authors list the phyla of organisms with which the negative interactions occur, but this could be improved by including the genera – which are what the figure actually shows.

On line 269-270 the authors state that “Pseudomonas was observed to interact negatively with Williamsia and this latter has an indirect link with Listeria” which seems to form the data on which the authors base their conclusion reported on lines 351-353 about Listeria forming niches within Pseudomonas biofilms. This is not intuitively obvious from Fig 4b as Pseudomonas does not appear in it and there are no observable links between Williamsia and Listeria in the figure. Thus, the conclusion needs more explanation to be believable.

Minor wording notes for the authors consideration

Line 70 “…independently of…” might be better as “…independent of…”

Line 363 “dissimilarity” should be ‘dissimilar”

Line 367 “…their niches are not common…” is unclear

Reviewer 3 Report

The manuscript presented by the authors is an interesting study on the co-occurrence of L. monocytogenes with other bacterial genera and bacterial diversity on cleaned conveyor surfaces in a swine slaughterhouse. The aim of the study was to describe the microbiota identified on the belt of three meat conveyors in a swine slaughterhouse after sanitation procedures to better understand the presence of L. monocytogenes on these surfaces. Both coltural (only for L. monocytogens) and molecular tools were used to characterize the microbial communities.

The overall quality of the manuscript is high. I have only three comments/suggestion:
- all smples were taken after cleaning and disinfection operations. Considering that this higly inluence the microbiota that is able to survive, I think that a description of the methodology and the materials used for the disinfection must be added
- even if it is clear that a good amount of work was done, the experimental design is composed by only four time of sampling. I think that the number of sampling should be increased if the authors have the opportunity. Otherwise, I suggest to avoid statements such as "From this background microbiota, a total of 6 genera, such as Sphingomonas, were found to interact negatively with Listeria spp., suggesting possible Listeria spp. growth inhibition, absence of shared habitats, or competition." (lines 390-392). This kind of conclusions are too generalistics, are not supported from a small amount of observations and requires in vitro challenge tests to be confirmed
- I suggest to consider to cite the following paper in the paper: Hadjicharalambous C, Grispoldi L, Goga BC. Quantitative risk assessment of Listeria monocytogenes in a traditional RTE product. EFSA J. 2019 Sep 17;17(Suppl 2):e170906. doi: 10.2903/j.efsa.2019.e170906. PMID: 32626464; PMCID: PMC7015512. 
